# Urolithin A Exhibits Antidepressant-like Effects by Modulating the AMPK/CREB/BDNF Pathway

**DOI:** 10.3390/nu17142294

**Published:** 2025-07-11

**Authors:** Yaqian Di, Rui Xue, Xia Li, Zijia Jin, Hanying Li, Lanrui Wu, Youzhi Zhang, Lei An

**Affiliations:** 1Key Laboratory of Geriatric Nutrition and Health, Beijing Technology and Business University, Ministry of Education, Beijing 100048, China; 2230302078@st.btbu.edu.cn (Y.D.); jinzijia2000@126.com (Z.J.); 2Beijing Key Laboratory of Neuropsychopharmacology, Beijing Institute of Pharmacology and Toxicology, Beijing 100850, China; hly19830718@sina.com (R.X.); lixia199156@163.com (X.L.); hanyingli199806@163.com (H.L.); wulanrui2023@163.com (L.W.)

**Keywords:** urolithin A, AMPK, CREB, depression, signaling pathway

## Abstract

Background/Objectives: Urolithin A (UA), a gut-derived metabolite of ellagitannins or ellagic acid, has recently gained attention for its potential benefits to brain health. The present research aimed to assess the antidepressant-like properties of UA in both in vitro and in vivo models and explored the molecular mechanisms underlying these effects. Methods: We investigated the antidepressant effects and mechanisms of UA in a model of corticosterone-induced damage to PC12 cells and in a model of chronic socially frustrating stress. Results: Our results demonstrate that UA treatment (5 and 10 μM) significantly alleviated cellular damage and inflammation in corticosterone (CORT)-treated PC12 cells. Furthermore, UA administration (50 and 100 mg/kg) significantly reduced immobility time in the mouse tail suspension test (TST) and forced swim test (FST), indicating its antidepressant-like activity. Additionally, treatment with UA led to the activation of the cAMP response element-binding protein (CREB)/brain-derived neurotrophic factor (BDNF) signaling cascade and triggered the activation of adenosine monophosphate-activated protein kinase (AMPK) during these processes. Importantly, pretreatment with AMPK-specific inhibitor Compound C abolished UA’s cytoprotective effects in PC12 cells, as well as its behavioral efficacy in the FST and TST, and its neurotrophic effects, highlighting the critical role of AMPK activation in mediating these effects. Furthermore, in the chronic social defeat stress (CSDS) mouse model, UA treatment (50 and 100 mg/kg) significantly alleviated depression-like behaviors, including reduced sucrose preference in the sucrose preference test, increased social avoidance behavior in the social interaction test, and anxiety-like behaviors, including diminished exploration, in the elevated plus maze test, suggesting the antidepressant-like and anxiolytic-like activities of UA. Moreover, UA treatment reversed elevated serum stress hormone levels, hippocampal inflammation, and the decreased AMPK/CREB/BDNF signaling pathway in the hippocampus of CSDS mice. Conclusions: Together, these results provide compelling evidence for UA as a viable dietary supplement or therapeutic option for managing depression.

## 1. Introduction

In recent years, depression has become a prevalent mental disorder globally, with an increasing economic burden of disability and suicide. Despite this escalating public health crisis, existing antidepressants frequently fail to meet clinical expectations regarding efficacy and safety. Most conventional antidepressants, such as selective serotonin reuptake inhibitors (SSRIs) and tricyclic antidepressants, primarily function through monoamine reuptake inhibition. By blocking the reuptake of neurotransmitters like serotonin, norepinephrine, or dopamine into presynaptic neurons, these drugs increase the concentration of monoamines in the synaptic cleft, thereby enhancing neurotransmission and alleviating depressive symptoms. However, despite their widespread use, these drugs are associated with various limitations, including delayed onset of action and a significant proportion of non-responsive patients. As a result, there is growing interest in discovering new, safe, and effective antidepressant compounds derived from natural products.

Produced by gut bacteria, Urolithin A (UA) serves as the major final metabolite derived from ellagitannins (ETs) and ellagic acid (EA), both of which are richly contained in numerous foods such as pomegranates, berries, and walnuts, and some medicinal plants [1]. In recent years, the interest in potential health benefits of UA for brain has been growing because of (1) its property of traversing the blood–brain barrier and accumulating in a targeted manner in the brain after absorption [2]; (2) supporting evidence from both preclinical and clinical studies that demonstrates UA’s benefits for brain health [3,4]; and (3) its potential to address challenges associated with microbiome-based treatments, including resistance to microbial colonization and differences in microbial composition among individuals [5]. These attributes position UA as a promising therapeutic agent for the central nervous system [6]. Substantial evidence suggests that consuming ET- or EA-rich foods, such as walnuts, berries, and pomegranates, can effectively alleviate depressive symptoms, reduce neuroinflammation, and enhance neuroprotective signaling [7,8,9]. Recently, Singh et al. reported that Urolithins (A and B) can selectively inhibit human monoamine oxidase A [10,11,12], indicating their potential efficacy in treating depression and anxiety.

A large number of studies have indicated that neurotropic mechanisms are tightly involved in brain health and disease [3,6,7]. Among various neurotrophins, cAMP response element-binding protein (CREB) and its downstream molecule brain-derived neurotrophic factor (BDNF) have been well documented in maintaining normal neuronal function such as synaptic plasticity, neurogenesis, and endogenous defense. As a transcription factor, the activity of CREB is regulated by several kinase pathways such as cAMP-dependent protein kinase A (PKA) and activated by the phosphorylation at serine 133 (Ser133). Numerous studies have shown that reduced p-CREB (Ser133) and BDNF expression are linked to depressive symptoms, and these downregulations can be reversed by antidepressants, which is thought to be related to their therapeutic effects. In a recent study, our results demonstrated that exposure to UA at concentrations of 5 μM and 10 μM led to a significant activation of the PKA/CREB/BDNF signaling pathway in H_2_O_2_-challenged SH-SY5Y cells [13]. Collectively, these findings suggest that UA may possess promising antidepressant-like activity.

As energy status in neurons and microglia is closely associated with brain functions, adenosine monophosphate-activated protein kinase (AMPK), a central regulator of energy and metabolism, has emerged as an important regulator in brain health and diseases. In recent decades, accumulating evidence has demonstrated that AMPK plays a significant role in the brain during aging, inflammation, and stress [13,14]. Abnormalities in AMPK activity disturb normal brain functions and compromise synaptic integrity. Several lines of research have shown that AMPK can regulate synaptic activation, neurogenesis, and endogenous defense via CREB signaling pathway [15,16]. In addition, UA has been evidenced as an AMPK activator in multiple types of cells (microglia, nucleus pulposus cells, and macrophages) [17,18,19] and tissues (brain, muscle, and pancreas) [17,20,21,22,23]. However, it is not yet fully understood how AMPK is involved in the health benefits of UA on the brain. Recently, increasing evidence supports AMPK as a novel target in the context of depression and antidepressant effects [13,15].

Therefore, this study aims to investigate the antidepressant-like activity of UA using cell and animal models, including corticosterone-treated PC12 cells, as well as behavioral despair models in C57BL/6J mice and chronic social defeat stress (CSDS) models in CD-1 mice. Additionally, we seek to elucidate the molecular mechanisms involved, particularly those associated with AMPK and the CREB-mediated neurotrophic signaling pathway.

## 2. Materials

Urolithin A (UA) was purchased from Nanjing Daosif Bio-technology Co., Ltd. (Nanjing, China) Duloxetine (DLX) was obtained from Zhejiang Liaoyuan Pharmaceutical Co., Ltd. (Taizhou, China). Polyoxyethylene castor oil was acquired from Beijing Fengyi Jingqiu Pharmaceutical Co (Beijing, China). Sigma-Aldrich (St. Louis, MO, USA) provided the following: corticosterone (CORT), promidium iodide (PI), 4′-6-diamidino-2-phenylindole (DAPI), and 4-(4,5-dimethyl-2-thiazolyl)-2,5-diphenyl-2H-tetrazolium bromide (MTT). Compound C (Dorsomorphin) was purchased from Shengshi Zhongfang (Beijing) Biotechnology Co., Ltd. (Shanghai, China). The following antibodies were used in Western blot: AMPK (2603), p-AMPK(Thr172) (2535), p-ACC (11818), ACC (3676), p-PKA substrate (4060), p-CREB (9198), CREB (9197), BDNF (47808), p-mTOR (5536), mTOR (2983), p-AKT (4060), and AKT (4691). These were all obtained from cell signal technology (CST). The adrenocorticotropic hormone (ACTH) ELISA kit (JL12373) and CORT ELISA kit (JL11918) were purchased from Jianglai Biological (Shanghai, China).

### 2.1. Cell Culture and Treatment

The ATCC (ATCC ^®^ CRL1721.1TM) provided rat pheochromocytoma (PC12) cells. PC12 cells were grown at 37 °C in a humidified environment with 95% air and 5% CO_2_ in RPMI 1640 media supplemented with 10% FBS, 5% HS, and 1% penicillin/streptomycin. Two to three days later, the culture medium was changed. In 96-well plates, PC12 cells were planted at a density of 2 × 10^4^ cells per well. To determine the optimal concentration of CORT or UA, the cells were subjected to varying concentrations of CORT (50–400 μM) or UA (0.1–20 μM) for 24 h. Cells were pretreated with several doses of UA (0.1–10 μM) for 12 h, and then exposed to 250 μM CORT for 24 h in order to determine the cytoprotective effects of UA in the CORT-induced PC12 cell model.

In the blocking tests, the PC12 cells were treated with Compound C (10 μM) for 3 h to block AMPK activation. And then PC12 cells were treated with either 5 or 10 μM of UA for 12 h and then exposed to 250 μM CORT for 24 h.

### 2.2. Cell Viability Assay

Cell viability was determined by MTT assay. Each well received 10 μL of the MTT chromogenic reagent with a concentration of 5 mg/mL, and then the plate was incubated at 37 °C for 4 h. After removing the supernatant, 100 μL of DMSO was added to each well to dissolve the formazan crystals, and the optical density (OD) was measured at a wavelength of 490 nm with a microplate reader. The results were reported as a percentage with respect to the control group, using the following formula: Cell viability (%) = (OD of treatment group − OD of blank group)/(OD of control group − OD of blank group) × 100%.

### 2.3. Reactive Oxygen Species Assay

The reactive oxygen species assay (ROS) was determined by the 2′,7′dichlorodihydrofuorescein diacetate (DCFH-DA) assay. PC12 cells were incubated with DCFH-DA, and then the plate was washed with PBS. The fluorescence intensity was measured using a fluorescence microplate reader at an excitation wavelength of 488 nm and an emission wavelength of 525 nm. Results were expressed as a percentage of the control group.

### 2.4. Determination of Apoptosis

Cell apoptosis was observed using DAPI/PI double staining under a fluorescence microscope. After treatment, PC12 cells were washed with 1 mL PBS and fixed with 4% paraformaldehyde for 15 min. The fixed cells were then washed again with 1 mL PBS and stained with 1 mL PI (5 μg/mL) solution and 1 mL DAPI (10 μg/mL) at room temperature for 10 min. Finally, the cells were washed with PBS and observed under a fluorescence microscope. Images were captured randomly from different focal areas. Untreated cells were used as the control.

### 2.5. Animals and Treatments

Seven-week-old male C57BL/6J mice, weighing between 18 and 22 g, were purchased from Beijing Huafukang Biotechnology Co., Ltd. (Beijing, China) CD-1 mice were obtained from Home-SPF (Beijing) Biotechnology Co., Ltd. (Beijing, China). The animals were housed in an environment with a 12 h light/dark cycle (light from 8:00 to 20:00), a temperature range of 22–24 °C, and a humidity level of 50–60%. Mice were kept in groups of 10 per cage and had ad libitum access to food and water throughout the study period. All animals were acclimatized for one week before the start of the experiment. To facilitate acclimatization, animals were transported to the experimental setting 24 h prior to the initiation of the study. All behavioral experiments were performed in a quiet, controlled environment. All animal experiments were carried out in compliance with the ARRIVE (Animal Research: Reporting of In Vivo Experiments) guidelines, which aim to improve the reporting quality of animal research. The study protocol was approved by the Institutional Animal Care and Use Committee of Beijing Institute of Pharmacology and Toxicology (Approval No. IACUC-DWZX-2023-582).

UA and DLX were dissolved in 5% castor oil, and the prepared solutions were given at a dosage volume of 20 mL per kilogram. In parallel, control mice received equivalent volumes of the solvent vehicle.

## 3. Experimental Design

The experimental scheme of the study is listed in Figure 1. The behavioral test order was carefully arranged to minimize carry-over effects and ensure accurate, reliable results. Tests were sequenced from more to less important, and from lower to higher stress, accurately reflecting the behavioral and physiological aspects of the mice.

### 3.1. Forced Swimming Test (FST)

Each mouse was separately put into a glass container containing 10 cm of water (at a temperature of 25 ± 1 °C) and forced to swim for 6 min [24]. The immobility time of each mouse was counted for the last 4 min.

### 3.2. Tail Suspension Test (TST)

The mice were suspended upside down by their tails with adhesive tape (approximately 2 cm from the tip of the tail) on a ledge, with their heads positioned 5 cm above the floor [25]. The recording time was 6 min, and the immobility time of each mouse was calculated for the last 4 min.

### 3.3. Locomotor Activity Test (LAT)

Mice were individually placed in a test box (25 cm × 25 cm × 25 cm), and the total traveled distance and duration of time spent in the central region were recorded for 10 min [26] using a video tracking system. Once each test was finished, 75% alcohol was used to clean the box to prevent odor interference.

### 3.4. Chronic Social Defeat Stress (CSDS) Model

#### 3.4.1. Screening Process for Aggressor CD-1 Mice

The screening process for aggressive CD-1 mice was performed according to a previously described method with modifications [27]. Retired male CD-1 mice were singly accommodated in cages, allowing them free access to food and water, and given seven days to acclimate. Seven-week-old C57BL/6J mice (used solely for screening) were employed to identify aggressive CD-1 mice. The screening was conducted in the resident CD-1 cages. C57BL/6J mice were introduced into the CD-1 cage for 180 s once daily over three consecutive days. To ensure consistency, the same CD-1 mouse was required to attack a different C57BL/6J mouse on each subsequent day. To meet the aggression criteria, a CD-1 mouse had to exhibit the following behaviors: (1) an attack latency of less than 30 s or at least two consecutive attacks on a C57BL/6J mouse entering the cage; (2) sustained aggression for at least two consecutive days, without human intervention during the screening process.

#### 3.4.2. CSDS Procedure

The defeat period lasted for 10 consecutive days. Briefly, C57BL/6J intruder mice were introduced into the resident cage of the aggressive CD-1 mice for 5–10 min each day. During this period, the aggressive CD-1 mice attacked the C57BL/6J intruders, prompting the intruder mice to display behaviors such as fear, submission, and avoidance. Following the social defeat, the intruder mice were separated by a divider and housed with the aggressor mice in the same compartment for 24 h. Behavioral experiments, including the sucrose preference test (SPT), social interaction test (SIT), and elevated plus maze test (EPMT), were conducted after the 10-day defeat period. Following the CSDS procedure, mice were classified as either susceptible or resilient based on their social interaction time. Given the primary objective of evaluating the antidepressant effects of UA, only susceptible mice were selected for subsequent experimental groups. This approach enabled us to focus on a population that consistently displayed depressive-like behaviors, allowing for a more accurate assessment of UA’s therapeutic effects.

### 3.5. Sucrose Preference Test (SPT)

Prior to the formal test, each mouse was provided with two bottles of distilled water for 24 h [28]. This was then replaced with one bottle of distilled water and one bottle of 1% sucrose solution for an additional 24 h, during which the mice were allowed free access to both fluids. On the third day (the formal SPT), the mice were given one bottle of distilled water and one bottle of 1% sucrose solution, with the positions of the two bottles switched after 12 h. The volumes of sucrose solution and distilled water consumed over a 24 h period were measured by weighing the bottles before and after the test. Sucrose preference (SP) was calculated as follows: SP (%) = (sucrose water intake/(sucrose water intake + distilled water intake)) × 100%.

### 3.6. Social Interaction Test (SIT)

The setup consisted of a white open arena for social interaction, with an empty wire mesh plexiglass enclosure placed within the arena [29]. The test was divided into two phases: the target-absent phase and the target-present phase, each lasting 150 s, separated by a 30 s break. During the target-absent phase, the C57BL/6J mice were allowed to habituate for 150 s. In the target-present phase, an unfamiliar CD-1 mouse was placed inside the plexiglass enclosure, and the C57BL/6J mice were then introduced into the arena and allowed to move freely for another 150 s. The time spent in the interaction zone during each phase was recorded separately. At the end of each trial, both the open arena and the plexiglass were cleaned with 75% alcohol.

### 3.7. Elevated Plus Maze Test (EPMT)

The EPM apparatus consisted of two open arms and two closed arms. Mice were gently placed in the central area of the maze, facing an open arm, and were allowed 5 min of free exploration [30]. The trials were recorded using VisuTrack XR-VT animal behavior analysis software (https://www.softmaze.com/s02/product/2021/03/10/2063962629.html, accessed on 6 May 2025, Shanghai Xinsoft Information Technology Co., Ltd., Shanghai, China), and exploratory behaviors were scored as follows: The percentage of time spent in the open arms was calculated as follows: (duration in open arms ÷ total exploration duration) × 100%. The percentage of time in the closed arms was determined by (time in closed arms ÷ total exploration time) × 100%. The percentage of entries into the open arms was computed as follows: (number of open-arm entries ÷ total number of arm entries) × 100%. The percentage of entries into the closed arms was computed as follows: (number of closed-arm entries ÷ total number of arm entries) × 100%.

### 3.8. Measurement of Adrenocorticotropic Hormone (ACTH) and Corticosterone (CORT) Levels

The levels of stress hormones, including ACTH and CORT, in the serum of the mice were measured using ELISA kits.

### 3.9. RT-qPCR

The RT-qPCR method was performed as described in our previous reports [13]. Briefly, total RNA was isolated from PC12 cells and hippocampal tissue using TriZol reagent. cDNA was synthesized using the ALL-in-One First-Strand Synthesis MasterMix. (LABLEAD, Beijing, China). Subsequently, PCR amplification was carried out on the cDNA using the Taq SYBR^®^ Green qPCR Premix kit (LABLEAD, Beijing, China), following the primer sequences generated. GAPDH was used as the internal reference gene, and the expression of the target gene was quantified with the application of the 2^−ΔΔCt^ formula. Table 1 shows the primer sequences that were utilized in this experiment.

### 3.10. Western Blot Analysis

Whole protein lysates were prepared using RIPA buffer or tissue lysis buffer containing inhibitors of protease and phosphatase. Protein concentrations were measured by means of the BCA method. Proteins with the same amount were separated through SDS-PAGE and transferred to PVDF membranes. The membranes were incubated with 5% skim milk for 1 h at room temperature to block nonspecific binding. The membranes were then incubated with primary antibodies maintained at 4 °C overnight. After the incubation was finished, ECL luminescent solution was applied for exposure. The imaging system was FluorChem (ProteinSimple, San Jose, CA, USA). Band intensities were quantified using the image analysis software ImageJ (version 1.8.0, developed by the National Institutes of Health, Bethesda, MD, USA). Protein expression levels were assessed by analyzing the gray values of the bands for p-ACC (1:1000), ACC (1:1000), p-AMPK (1:1000), AMPK (1:1000), p-PKA substrate (1:1000), p-CREB (1:1000), CREB (1:1000), BDNF (1:1000), p-mTOR (1:1000), mTOR (1:1000), p-AKT (1:1000), AKT (1:1000), and β-actin (1:1000) primary antibodies.

### 3.11. Statistical Analysis

The experimental data are presented as mean ± SEM and were statistically analyzed using GraphPad Prism 9.0.0 software. One-way analysis of variance (ANOVA), followed by Dunnett’s *t*-test, was performed for group comparisons, and Student’s *t*-test was used to assess differences between two groups. Statistical significance was defined as a *p*-value less than 0.05.

## 4. Results

### 4.1. UA Inhibited CORT-Induced Damage in PC12 Cells

As shown in Figure 2A,B, we chose CORT at a dose of 250 μM for modeling, which led to a decrease in cell viability, such that it reached approximately 75% of the control level, and chose UA at concentrations of 0.1 to 10 μM for further research, which had no impact on cell viability. As illustrated in Figure 2C, UA (0.1–10 μM) incubation significantly restored the decreased cell viability induced by CORT (*p* < 0.001). Moreover, as shown in Figure 2D, UA (5 and 10 μM) treatment significantly alleviated the excessive generation of ROS triggered by CORT (*p* < 0.001).

Additionally, as shown in Figure 2E, cells that exhibit bright red fluorescence with PI staining are identified as necrotic, while those showing light blue (living) or bright blue (early apoptosis) with DAPI staining, and light red (late apoptosis) with PI staining are considered to be undergoing apoptosis at different stages. Compared with the control, CORT-treated cells exhibited bright blue and bright red fluorescence, indicating that CORT induced significant cell apoptosis and necrosis. In contrast, UA (5 and 10 μM) significantly attenuated the bright blue and bright red fluorescence, suggesting that UA reversed CORT-induced cell apoptosis. These results further confirm that UA can relieve CORT-induced damage in PC12 cells.

As shown in Figure 2F, CORT treatment induced significant elevation in the levels of pro-inflammatory cytokines including IL-1β, IL-6, and TNF-α (*p* < 0.001), whereas UA (5 and 10 μM) treatment significantly restored the increased levels of these pro-inflammatory cytokines (*p* < 0.001). These results indicate that UA can inhibit CORT-induced inflammation in PC12 cells, which may underlie the cytoprotective role of UA in CORT treated PC12 cells.

### 4.2. UA Upregulated AMPK Activity and CREB-Mediated Neurotrophic Signaling in CORT-Treated PC12 Cells

Then, we further explored the impact of UA on AMPK activity and CREB-mediated neurotrophic signaling pathway in CORT-treated PC12 cells. As shown in Figure 3A–E, treatment with CORT (250 μM) significantly reduced the phosphorylation level of AMPK (Thr172), along with the phosphorylation level of its downstream target protein ACC (Ser79), which represents AMPK activity, whereas treatment with UA at concentrations of 5 and 10 μM significantly reversed the decreased expression of p-AMPK and p-ACC (*p* < 0.05 or *p* < 0.001), leading a dramatic increase in AMPK activity. The results are consistent with previous reports that UA is an AMPK activator [1].

PKA, as the major phosphokinase of CREB, induces the phosphorylation of CREB at the Ser133 site, which subsequently activates CREB-mediated transcription, such as that of BDNF. In this study, we used the p-PKA substrate antibody to measure PKA activity. As shown in Figure 3G–J, UA (5 and 10 μM) treatment significantly restored the decreased expression of p-PKA substrates, as well as the decreased expression of p-CREB and BDNF in CORT-treated PC12 cells (*p* < 0.001). These results suggest UA can significantly upregulate CREB-mediated neurotrophic signaling, which is in line with our previous findings in H_2_O_2_-treated SH-SY5Y cells [13].

### 4.3. Blocking AMPK Activity Abolished the Cytoprotective and Neurotrophic Effects of UA in CORT-Treated PC12 Cells

As several lines of research have shown that AMPK can regulate neuroplasticity such as synaptic activation, neurogenesis, and endogenous defense through the CREB-mediated neurotrophic signaling pathway [14,15], we determined how AMPK was involved in the cytoprotective and neurotrophic effects of UA in CORT-treated PC12 cells. We used the AMPK specific inhibitor Compound C (10 μM) to pretreat PC12 cells prior to UA. As shown in Figure 4A,B, treatment with Compound C alone did not affect the cell viability or the expression of pro-inflammatory cytokines. However, pretreatment with Compound C abolished the cytoprotective and anti-inflammatory effects of UA in CORT-treated PC12 cells. Similarly, Western blot analysis results (Figure 4C–K) demonstrate that in the presence of Compound C, both the activation of AMPK and the upregulation of CREB-mediated signaling by UA were abolished. These data indicate that AMPK activation is essential for both the cytoprotective and neurotrophic effects of UA in CORT-treated PC12 cells.

### 4.4. UA Produced Antidepressant-like Effects in the FST and TST in Mice

As shown in Figure 5A,B, compared with the control group, UA administration at 50 or 100 mg/kg significantly decreased the immobility time in both the TST and FST in mice (*p* < 0.05), with effects similar to those observed with DLX. Moreover, UA (50 and 100 mg/kg) administration did not affect the total distance traveled and the time spent in the central area in the OFT (Figure 5C,D), indicating that UA has no excitatory or inhibitory effects on the central nervous system, thus excluding the false-positive results in the TST and FST. These findings suggest that UA exhibits antidepressant-like effects in behavioral despair models.

### 4.5. UA Increased AMPK Activity and CREB-Mediated Neurotrophic Signaling in the Mouse Hippocampus

As shown in Figure 6A–F, UA (50 and 100 mg/kg) administration significantly increased the expression of p-AMPK, p-CREB, and BDNF in the mouse hippocampus, consistent with the results observed in PC12 cells. These findings suggest that UA activates AMPK and upregulates the CREB/BDNF neurotrophic signaling pathway in vivo, which may underlie the antidepressant-like effects of UA observed in the TST and FST.

### 4.6. Activation of AMPK Is Necessary for the Antidepressant Effects of UA in the TST and FST

To further investigate the role of AMPK in the antidepressant-like effects of UA, we pretreated mice with Compound C (1.45 mg/kg) 30 min before the behavioral tests. The TST and FST were then conducted. As shown in Figure 7A,B, Compound C treatment alone did not affect immobility time in either the TST or FST. However, Compound C pretreatment abolished the antidepressant-like effects of UA (50 mg/kg) in both tests. In contrast, Compound C pretreatment did not significantly affect the effects of DLX in the TST or FST. Additionally, as shown in Figure 7C,D, in the LAT, there were no significant differences in total distance traveled or time spent in the central area among groups, suggesting no central excitatory or inhibitory effects of any treatment. These findings indicate that AMPK activation is required for the antidepressant effects of UA, but not for those of DLX, in the TST and FST.

### 4.7. Activation of AMPK Is Necessary for the Neurotrophic Effects of UA in the Hippocampus of Mice

Furthermore, we examined how AMPK was involved in the neurotrophic effects of UA in mice. As shown in Figure 8, Compound C pretreatment (1.45 mg/kg) abolished both the activation of AMPK and the upregulation of CREB-mediated signaling by UA. These findings suggest that AMPK activation is required for both the antidepressant-like effects of UA in behavioral despair models and its neurotrophic effects in the hippocampus.

### 4.8. UA Alleviated the CSDS-Induced Depression-like and Anxiety-like Behaviors in Mice

As shown in Figure 9C, in the SPT, the sucrose preference of mice in the CSDS group was significantly reduced compared to the control group, indicating the presence of anhedonia. However, the administration of UA (50 and 100 mg/kg) significantly reversed the reduced sucrose preference, suggesting that UA can alleviate anhedonia symptoms in CSDS mice. In the SIT, as shown in Figure 9D, there were no differences in the time spent in the interaction zone during the target-absent phase. However, during the target-present phase, the CSDS group spent less time in the interaction zone compared to the control group, indicating social avoidance behavior. The administration of 100 mg/kg of UA led to a significant increase in the time spent in the interaction zone, suggesting that UA can alleviate social avoidance behavior in CSDS mice. In the EPMT, as shown in Figure 9F, the CSDS group exhibited a significant decrease in both the time spent in the open arms and the number of entries into the open arms compared to the control group. However, treatment with UA (50 and 100 mg/kg) markedly reversed these changes (*p* < 0.05 or *p* < 0.01). Additionally, UA treatment significantly reduced the time spent in the closed arms and the number of entries into the closed arms in CSDS mice, suggesting the anxiolytic effect of UA. These findings suggest that UA can alleviate depression-like and anxiety-like behaviors in CSDS mice. Treatment with DLX (10 mg/kg) produced similar effects.

### 4.9. UA Attenuated the CSDS-Induced Oversecretion of Serum Stress Hormones and Hippocampal Inflammation

As shown in Figure 10A,B, CSDS induced a significant increase in serum stress hormones, including ACTH and CORT, in C57BL/6J mice. However, the administration of UA (50 and 100 mg/kg) and DLX (10 mg/kg) significantly reduced ACTH and CORT levels (*p* < 0.05 or *p* < 0.01). Additionally, treatment with UA (50 mg/kg) and DLX (10 mg/kg) significantly inhibited the oversecretion of pro-inflammatory cytokines, including IL-6 and IL-1β, in the hippocampus of CSDS mice (*p* < 0.05). Furthermore, we assessed the effects of UA on the AKT/mTOR pathway, which is closely linked to stress and inflammation signaling [24]. As shown in Figure 10E–I UA treatment significantly decreased the CSDS-induced elevation of p-mTOR and p-AKT. These findings suggest that UA can reverse the CSDS-induced increase in stress hormones and hippocampal inflammation, which may contribute to its antidepressant-like effects in CSDS mice.

### 4.10. UA Reversed the CSDS-Induced AMPK/CREB/BDNF Signaling Downregulation in the Hippocampus

Additionally, we investigated if the AMPK/CREB/BDNF signaling pathway participates in the antidepressant-like effects of UA in the CSDS model. As shown in Figure 11B,C, the expression of p-AMPK, p-CREB, and BDNF was significantly reduced in the hippocampus of CSDS mice. However, treatment with UA (50 and 100 mg/kg) significantly reversed these changes (*p* < 0.05 or *p* < 0.01), comparable to the impact of DLX. Moreover, the upregulation of CREB/BDNF signaling by UA was even stronger than that induced by DLX (*p* < 0.05), indicating that UA exerts a notable neurotrophic effect in the hippocampus of CSDS mice. These findings further suggest that the upregulation of the AMPK/CREB/BDNF signaling pathway may be a key mechanism underlying the antidepressant-like effects of UA.

## 5. Discussion and Conclusions

To sum up, the findings of the current research demonstrate the following: (1) The treatment with UA notably counteracted CORT-induced damage and inflammation, as well as the decrease in AMPK and CREB signaling pathway in PC12 cells, with AMPK activation being essential for its cytoprotective and neurotrophic effects. (2) UA treatment demonstrated significant antidepressant-like effects in the mouse TST and FST and increased the AMPK/CREB/BDNF signaling pathway in the hippocampus, with AMPK activation also playing a crucial role in its behavioral efficacy and neurotrophic effect. (3) In the CSDS model, UA treatment produced both antidepressant-like and anxiolytic-like effects, reversed elevated serum stress hormone levels and hippocampal inflammation, and restored the AMPK/CREB/BDNF signaling pathway.

Although evidence shows that consuming ET- or EA-rich food can reduce the incidence of depression and improve mood [31], elucidating the mechanisms underlying these effects remains challenging. Because EA and ETs have very low bioavailability in vivo, they cannot even pass through the blood–brain barrier [32]. Our study uniquely provides direct, multi-model evidence of UA’s antidepressant effects and clarifies its activation of the AMPK/CREB/BDNF pathway through causal experiments, suggesting that UA is responsible for the antidepressant effects of food rich in ETs and EA. Our work provides a more comprehensive understanding of UA’s potential therapeutic application, filling a significant gap in existing research on UA’s antidepressant mechanisms. Conventional antidepressants like SSRIs primarily target neurotransmitter reuptake, while UA activates the AMPK/CREB/BDNF pathway, promoting long-term neuroplasticity. This distinct mechanism may offer more sustainable antidepressant effects and reduce side-effects associated with neurotransmitter-only modulation. Additionally, to further validate the statistical power of our study, we conducted post hoc power analyses for the key outcomes. For example, the post hoc power analysis of the FST results of mice after UA (50 mg/kg) intervention showed that with a sample size of 10 mice per group, an observed effect size of 0.6, and a significance level of α = 0.05, the statistical power reached 82%. This indicates that, given the current sample size and effect size, we have an 82% probability of detecting the observed effect of UA on immobility time in the FST, suggesting that the research results are highly reliable.

The sucrose preference test is a classic assessment for anhedonia in rodents, one of the hallmark symptoms of depression. In CSDS model, UA treatment demonstrated a more potent effect in enhancing sucrose preference compared to the positive control drug DLX. Moreover, in the SIT, UA treatment at a dose of 100 mg/kg significantly increased the time spent in the interaction zone, whereas DLX did not. These findings underscore the advantages of UA in alleviating anhedonia and reducing social avoidance behavior. Additionally, CREB-mediated synaptic and neuronal changes in hippocampus and prefrontal cortex help modulate the brain’s stress response and regulate neurotransmitter systems, ultimately reducing depression-like and anxiety-like behaviors. It is worth noting that UA treatment produced notable neurotrophic effects in both CORT-treated PC12 cells and the hippocampus of normal and CSDS mice, even stronger than those of DLX, suggesting UA may be a neurotrophic agent, and upregulating CREB-mediated neurotrophic signaling may be a crucial mechanism underlying its effects.

Recent studies provide compelling evidence that AMPK may serve as a novel target for antidepressant therapies. Chronic stress, the most classical inducement of depression, has been shown to lead to AMPK inactivation [14]. Conversely, AMPK activation mediates the antidepressant-like effects of various treatments, including conventional antidepressants and aerobic exercise [24,25,26,27]. However, the precise mechanisms by which AMPK exerts these effects remain to be fully elucidated. In the present study, pretreatment with AMPK-specific inhibitor Compound C abolished both UA’s cytoprotective and neurotrophic effects in PC12 cells, as well as the antidepressant-like and neurotrophic effects in mice. These results not only highlight that AMPK activation is a crucial mechanism underlying the antidepressant-like effects of UA but also suggest that the metabolic regulation of neurotrophic signaling may be a pivotal mechanism through which AMPK modulates brain function. While Compound C is commonly used to study AMPK-dependent pathways, we acknowledge its potential off-target effects. Although a relatively low dose (10 µM) was employed to minimize nonspecific binding, potential off-target interactions with kinases such as CDK1 or GSK3β cannot be completely excluded. This underscores the need for future validation using genetic models or more selective inhibitors. In addition, in the behavioral despair model, 50 mg/kg of UA significantly reduced the immobility time in the mouse TST and FST and simultaneously increased p-AMPK and BDNF expression in the hippocampus. In the CSCD model, the same dose of UA notably enhanced the SP in mice and upregulated BDNF expression in the hippocampus. These results clearly demonstrate the consistency between the molecular mechanisms and behavioral effects of UA.

Although this study provides compelling preclinical evidence for the antidepressant effects of UA in animal models, caution should be exercised when considering the translation of these findings to human use. Firstly, the bioavailability of UA in humans differs significantly from that in animal models. After oral administration in humans, UA undergoes a complex process of intestinal metabolism, and the amount that actually enters the bloodstream and reaches the brain may be much lower than the effective dose observed in animal experiments [33]. Secondly, the effective concentration of UA in the brain is a crucial determinant of its antidepressant effects. Due to the presence of the blood–brain barrier, how to ensure that UA reaches specific brain regions at sufficient concentrations to regulate neurotransmitters and neural plasticity requires further investigation. Additionally, differences in the gut microbiota among individuals may lead to variations in the efficiency of ET metabolism into UA [34]. This inter-individual variability in metabolism will directly affect the final in vivo concentration of UA and its antidepressant efficacy. Future research should conduct human clinical trials and incorporate pharmacokinetic and metabolomics analyses to systematically evaluate the safety, efficacy, and optimal dosage of UA in humans [35]. This study provides compelling evidence supporting UA as a promising dietary supplement or therapeutic agent for depression.

## Figures and Tables

**Figure 1 nutrients-17-02294-f001:**
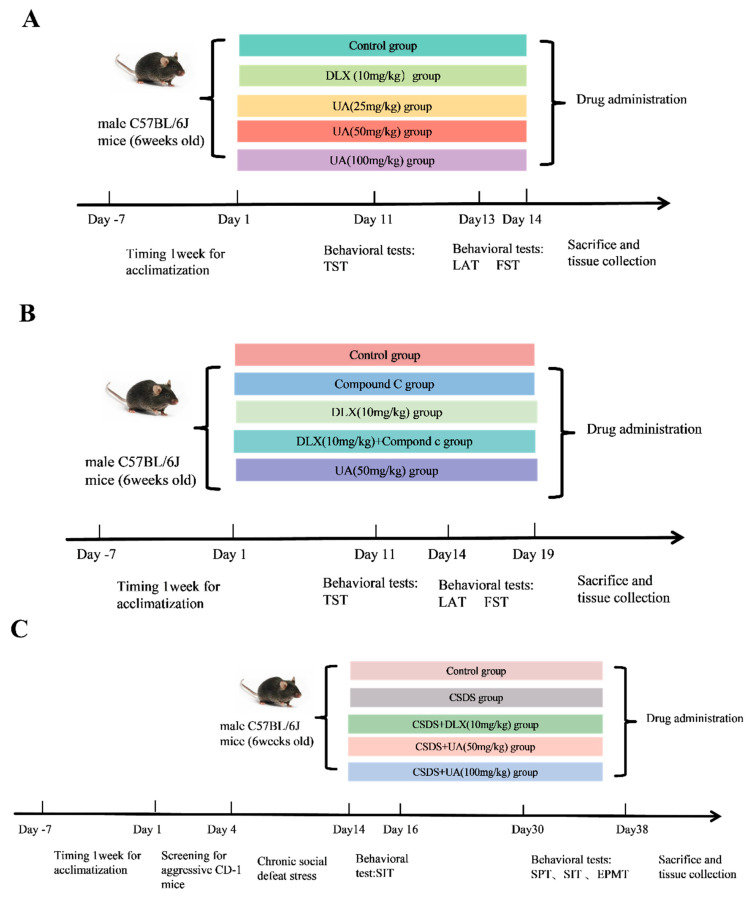
Experimental scheme of the study. (**A**) The experimental procedure of behavioral despair models. For a period of 14 days, the mice were subjected to treatment with the solvent vehicle, DLX at a dose of 10 mg/kg or UA at doses of 25, 50, and 100 mg/kg. The behavioral tests, including the tail suspension test (TST), locomotor activity test (LAT), and forced swimming test (FST), were performed 1 h after the treatment on the 11th, 13th and 14th day, respectively. The sacrifice and tissue collection were performed on the 14th day. (**B**) The experimental procedure of AMPK blocking in behavioral despair models. The mice were treated with the solvent vehicle, DLX (10 mg/kg) or UA (50 and 100 mg/kg), for 19 days. The mice were injected with AMPK inhibitor Compound C (1.45 mg/kg) half an hour before the usual treatment, and 1 h after the treatment; the behavioral tests, including the TST, LAT, and FST, were performed on the 11th, 14th, and 19th day, respectively. The sacrifice and tissue collection were performed on the 19th day. (**C**) The experimental procedure of chronic social defeat stress (CSDS). The defeat period lasted for 10 days. Then, after 2 days of SIT screening, the CSDS mice were treated with the solvent vehicle, DLX (10 mg/kg) or UA (50 and 100 mg/kg), for 14 days. The behavioral tests, including the sucrose preference test (SPT), social interaction test (SIT), and elevated plus maze test (EPMT), were performed 1 h after the treatment on the 30th, 32nd, and 36th day. The sacrifice and tissue collection were performed on the 38th day.

**Figure 2 nutrients-17-02294-f002:**
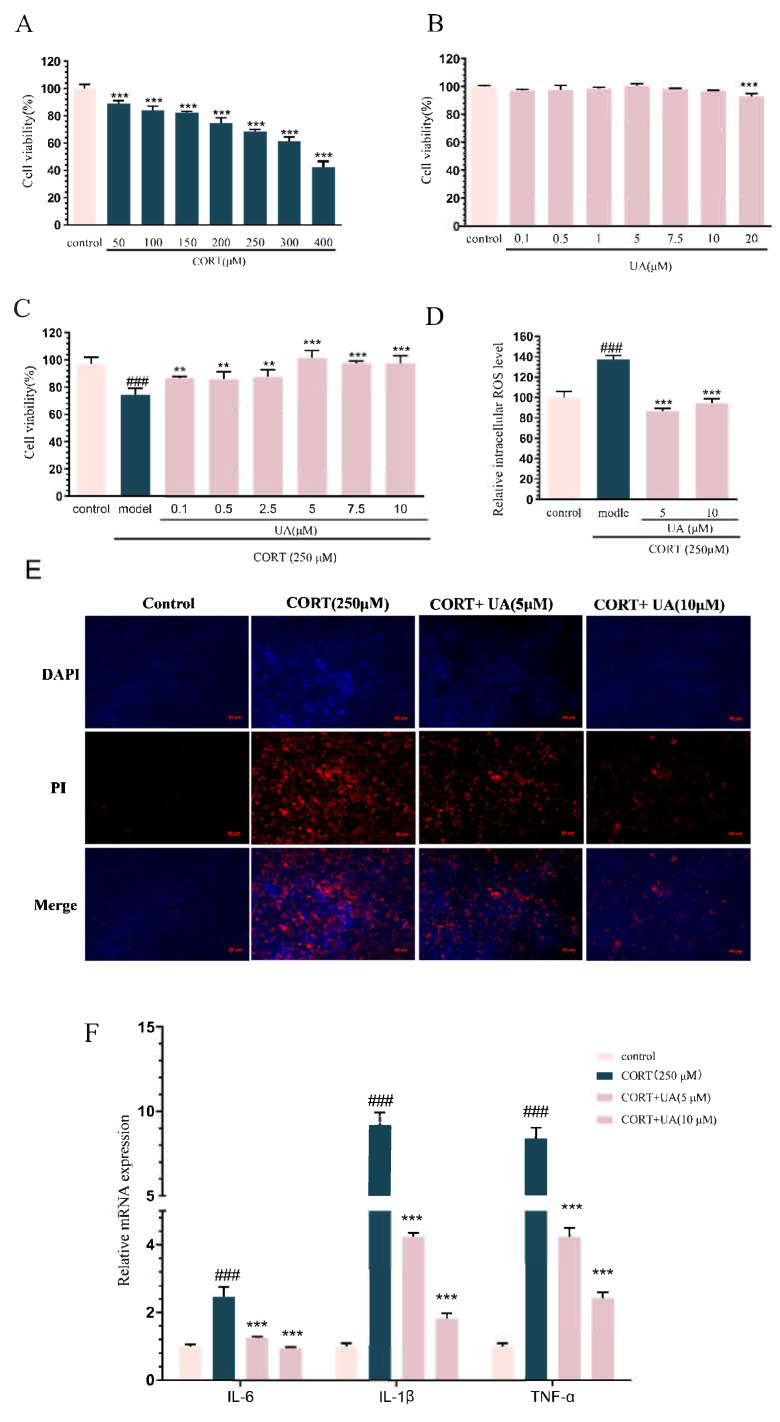
UA inhibited CORT-induced damage in PC12 cells. (**A**,**B**) The cell viability of PC12 cells exposed to varying doses of CORT (50–400 μM) or UA (0.1–20 μM) for a 24 h period. (**C**–**F**) The cell viability, the percentage of ROS levels, the cell apoptosis (bright blue fluorescence indicates early-stage apoptotic cells, as labeled by DAPI, while bright red fluorescence represents necrotic cells, which are labeled by PI due to their damaged cell membranes), and the gene expression of pro-inflammatory factors in PC12 cells exposed to UA (0.1–10 μM) for 12 h and then for a further 24 h period with CORT (250 μM). Each column represents mean ± SEM, *n* = 6. The results are representative of at least three experiments. ### *p* < 0.001 compared with the control group; ** *p* < 0.01 and *** *p* < 0.001 compared with the control group or the CORT-treated group.

**Figure 3 nutrients-17-02294-f003:**
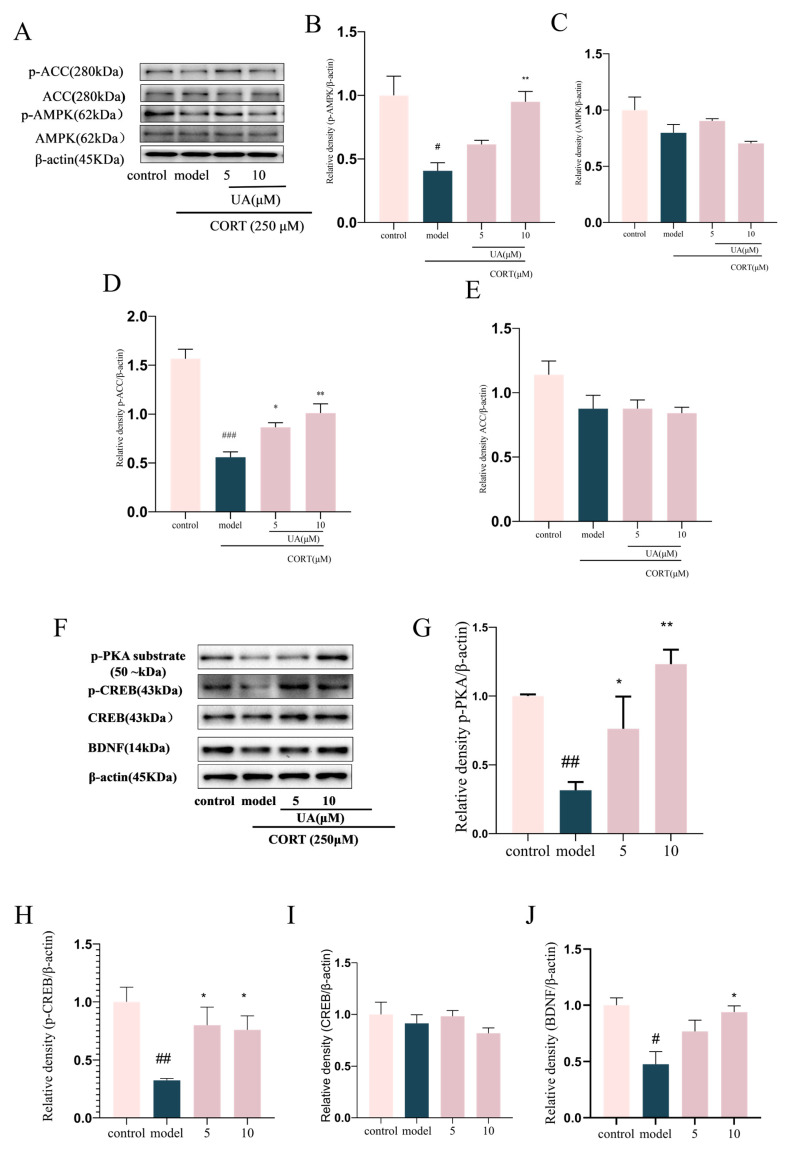
UA upregulated AMPK activity and CREB-mediated neurotrophic signaling in CORT-treated PC12 cells. Cells were cultivated in the presence of UA at concentrations of 5 and 10 μM for a duration of 12 h, following which they were subjected to treatment with CORT (250 μM) for another 24 h. (**A**–**J**) Representative Western blot and quantitative analysis of p-ACC, ACC, p-AMPK, AMPK, p-PKA substrate, p-CREB, CREB, and BDNF expression. Each column represents mean ± SEM, *n* = 3. The results are representative of at least three experiments. # *p* < 0.05, ## *p* < 0.01 and ### *p* < 0.001 compared with the control group; * *p* < 0.05 and ** *p* < 0.01 compared with the CORT-treated group.

**Figure 4 nutrients-17-02294-f004:**
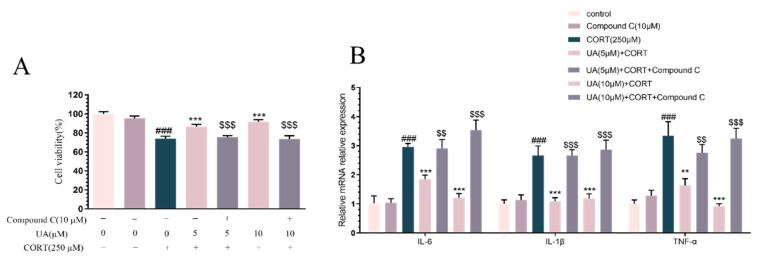
Blocking AMPK activity abolished the cytoprotective and neurotrophic effects of UA in CORT-treated PC12 cells. Cells were preincubated with Compound C (10 μM) for 3 h; subsequently, the cells were exposed to UA at concentrations of 5 and 10 μM for a duration of 12 h, and CORT (250 μM) for another 24 h. (**A**,**B**) The cell viability and the gene expression of pro-inflammatory factors of PC12 cells were detected. Each column represents mean ± SEM, *n* = 6. (**C**–**K**) Representative Western blot and quantitative analyses of p-ACC, ACC, p-AMPK, AMPK, p-PKA, p-CREB, CREB, and BDNF expression in PC12 cells. Each column represents mean ± SEM, *n* = 3. The results are representative of at least three experiments. ### *p* < 0.001 compared with the control group; ** *p* < 0.01 and *** *p* < 0.001 compared with the CORT-treated group; $ *p* < 0.05, $$ *p* < 0.01, and $$$ *p* < 0.001 compared with the respective non-Compound C group.

**Figure 5 nutrients-17-02294-f005:**
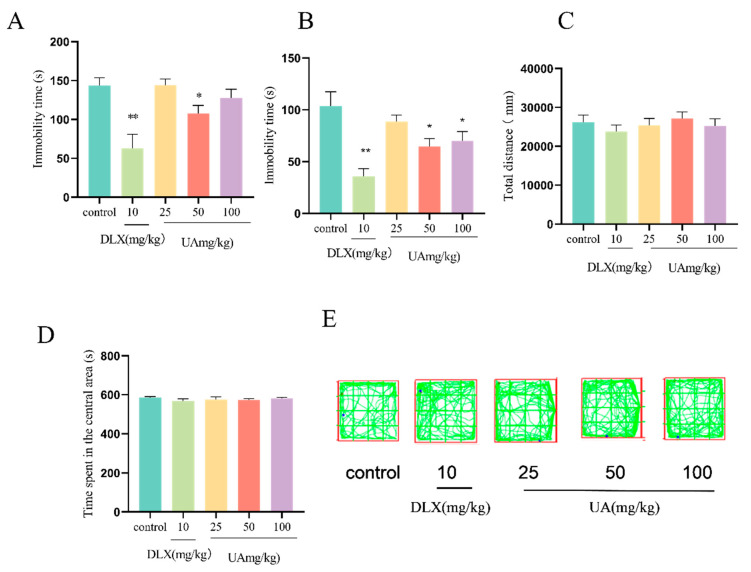
UA produced antidepressant-like effects in the TST and FST in mice. (**A**) TST, (**B**) FST, and (**C**–**E**) total distance, time spent in the central area, and representative track image of total movement in LAT. Each column represents mean ± SEM, *n* = 10. * *p* < 0.05 and ** *p* < 0.01 compared with the control group.

**Figure 6 nutrients-17-02294-f006:**
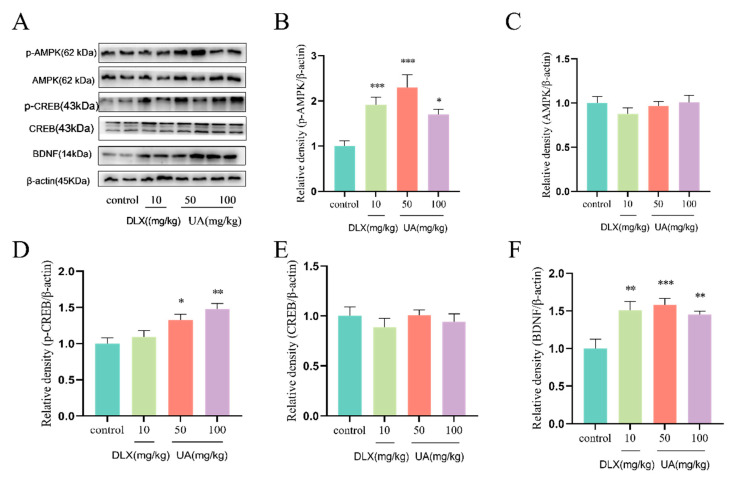
UA increased AMPK activity and CREB-mediated neurotrophic signaling in mouse hippocampus. (**A**–**F**) Representative Western blot and quantitative analysis of p-AMPK, AMPK, p-CREB, CREB, and BDNF expression. Each column represents mean ± SEM, *n* = 3. The results are representative of at least three experiments. * *p* < 0.05, ** *p* < 0.01, and *** *p* < 0.001 compared with the control group.

**Figure 7 nutrients-17-02294-f007:**
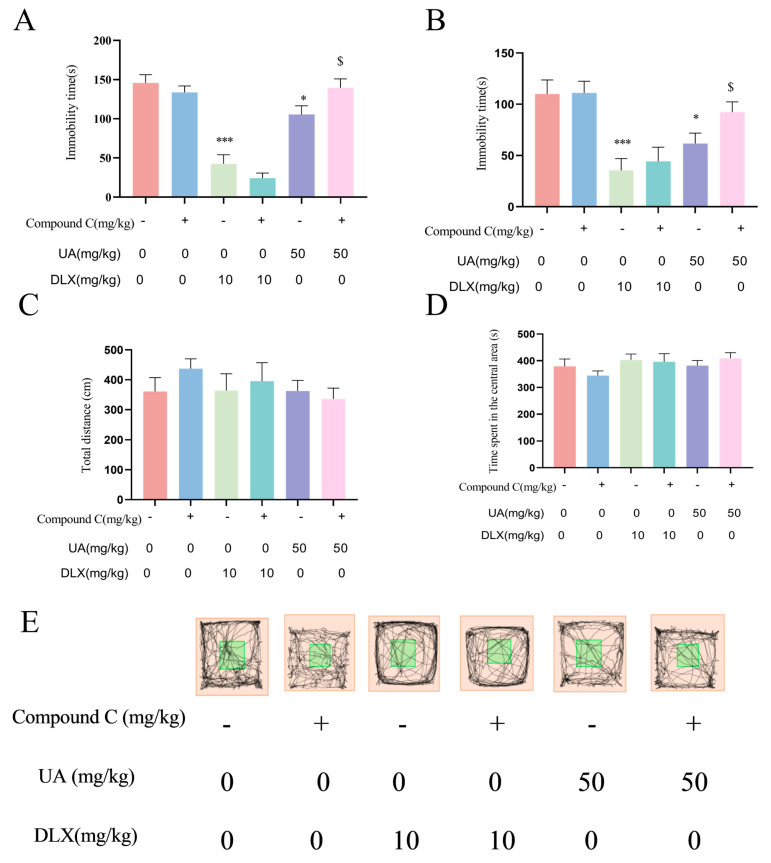
Activation of AMPK is necessary for the antidepressant effects of UA in the TST and FST. (**A**) TST. (**B**) FST. (**C**–**E**) Total distance, time spent in the central arear, and representative track image in LAT. Each column represents mean ± SEM, *n* = 10. * *p* < 0.05 and *** *p* < 0.001 compared with the control group; $ *p* < 0.05 compared with the respective non-Compound C group.

**Figure 8 nutrients-17-02294-f008:**
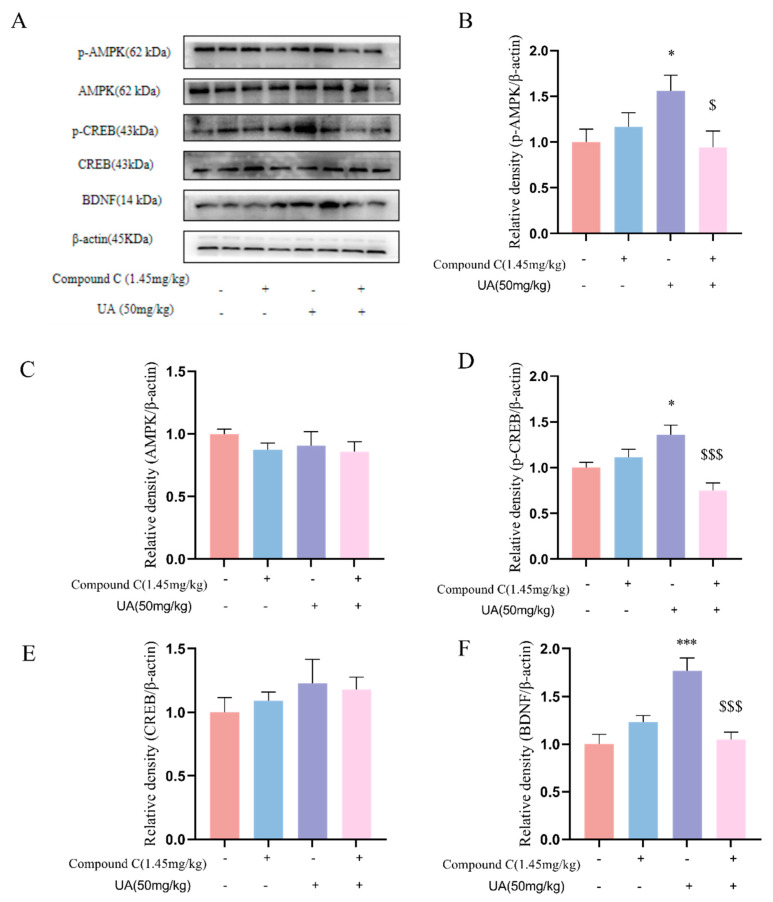
Activation of AMPK is necessary for the neurotrophic effects of UA in the mouse hippocampus (**A**–**F**). Representative Western blot and quantitative analysis of p-AMPK, AMPK, p-CREB, CREB, and BDNF expression. Each column represents mean ± SEM, *n* = 3. The results are representative of at least three experiments. * *p* < 0.05 and *** *p* < 0.001 compared with the control group; $ *p* < 0.05 and $$$ *p* < 0.001 compared with the respective non-Compound C group.

**Figure 9 nutrients-17-02294-f009:**
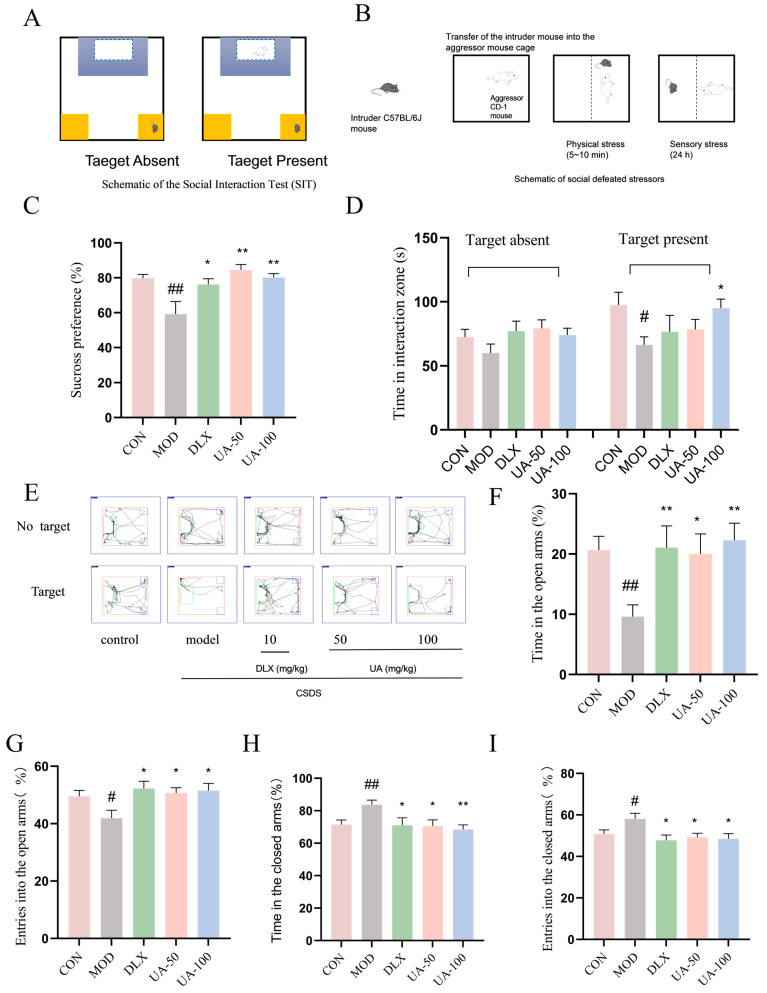
UA alleviated CSDS-induced depression-like and anxiety-like behaviors in mice. (**A**,**B**) Schematic representation of the CSDS procedure, (**C**) the sucrose preference in the SPT, (**D**) the time spent in the interaction zone during target-absent phase and target-present phase in the SIT, (**E**) representative track image of total movement in the SIT, (**F**–**I**) the time spent in the open arms (the closed arms) and the number of entries into the open arms (the closed arms) in the EPMT. Each column represents mean ± SEM, n = 10. # *p* < 0.05 and ## *p* < 0.01 compared with the control group; * *p* < 0.05 and ** *p* < 0.01 compared with the CSDS group.

**Figure 10 nutrients-17-02294-f010:**
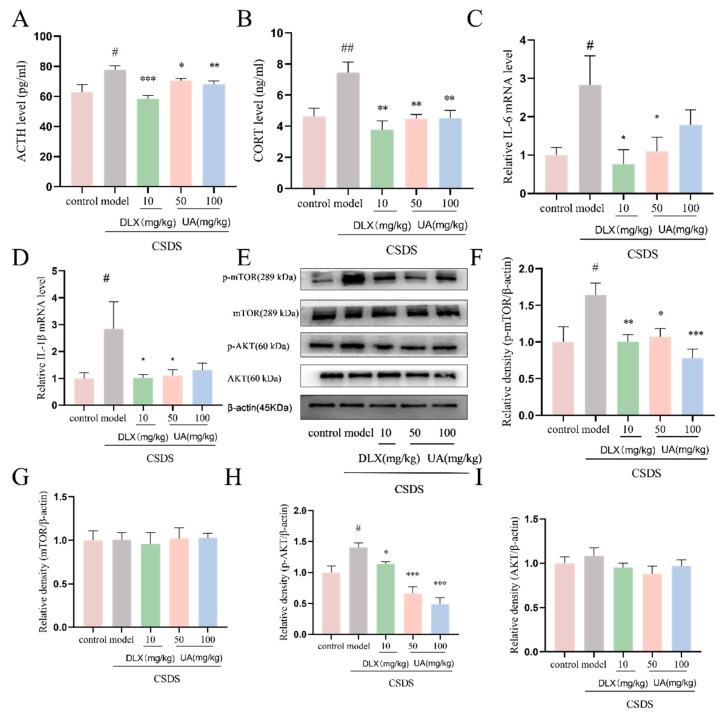
UA attenuated CSDS-induced oversecretion of serum stress hormones and hippocampal inflammation. (**A**) ACTH level. (**B**) CORT level. Each sample was assayed in duplicate, and absorbance values were recorded using a microplate reader. (**C**,**D**) The genes expression of pro-inflammatory factors in CSDS-induced detected by RT-qPCR. Each column represents mean ± SEM, n = 10. (**E**–**I**) Representative Western blot and quantitative analysis of p-mtor, mTOR, p-AKT, and AKT expression. Each column represents mean ± SEM, n = 3. The results are representative of at least three experiments. # *p* < 0.05 and ## *p* < 0.01 compared with the control group; * *p* < 0.05, ** *p* < 0.01 and *** *p* < 0.001 compared with the CSDS group.

**Figure 11 nutrients-17-02294-f011:**
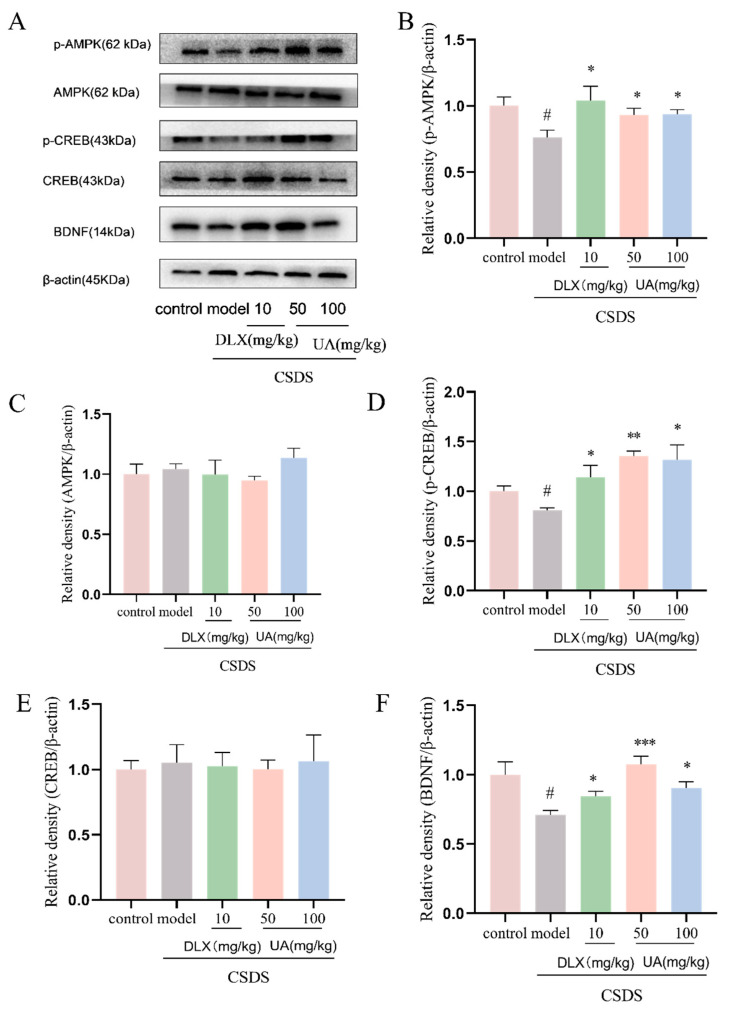
UA reversed the CSDS-induced AMPK/CREB/BDNF signaling downregulation in the hippocampus. (**A**–**F**) Representative Western blot and quantitative analyses of p-AMPK, AMPK, p-CREB, CREB, and BDNF expression. Each column represents mean ± SEM, n = 3. The results are representative of at least three experiments. # *p* < 0.05 compared with the control group; * *p* < 0.05, ** *p* < 0.01, and *** *p* < 0.001 compared with the CSDS group.

**Table 1 nutrients-17-02294-t001:** Primer sequences of qPCR.

Gene	Primer	Sequence
IL-6	forward	5′-ACATCGACCCGTCCACAGTAT-3′
IL-6	reverse	5′-AGTGGTATAGACAGGTCTGTTGG-3′
IL-1β	forward	5′-GAAATGCCACCTTTTGACAGTG-3′
IL-1β	reverse	5′-TGGATGCTCTCATCAGGACAG-3′
TNF-α	forward	5′-CAGGCGGTGCCTATGTCTC-3′
TNF-α	reverse	5′-CGATCACCCCGAAGTTCAGTAG-3′
GAPDH	forward	5′-AGCCTCGTCCCGTAGACAAAA-3′
GAPDH	reverse	5′-TGGCAACAATCTCCACTTTGC-3′

## Data Availability

The original contributions presented in this study are included in the article. Further inquiries can be directed to the corresponding authors.

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
