# Peer review of "Urolithin A Exhibits Antidepressant-like Effects by Modulating the AMPK/CREB/BDNF Pathway"

_nutrients, 2025, doi:10.3390/nu17142294_

Round 1

Reviewer 1 Report

Comments and Suggestions for Authors

In the manuscript, the authors evaluated the antidepressant-like effects of Urolithin A (UA) in both in vitro and in vivo models and explored the molecular mechanisms underlying these effects. In addition to the behavioral antidepressant-like effects of UA, the Authors found UA treatment upregulating the cAMP response element-binding protein (CREB)/brain-derived neurotrophic factor (BDNF) signaling pathway and inducing an activation of adenosine monophosphate-activated protein kinase (AMPK) during these processes.

The research is original, the authors present the problem clearly, and provide satisfactory background information. However, the manuscript needs improvement according to the comments below:

  1. Although the authors explored some intracellular signaling pathways, the manuscript remains descriptive and lacks sufficient discussion of the mechanisms that may be involved in the antidepressant-like effects of UA. For example, which observed effects induced by UA may relate to its MAO inhibitor activity, etc…
  2. Methods: Please provide more details when describing the Western blot analysis, details about the software, the manufacturer, the place of origin, etc.
  3. Results: There is a serious problem with Fig. 2, overlapping images, which are entirely illegible.
  4. Results: The authors declare that they measured PKA activity, yet they describe only p-PKA assessment without providing the value for unphosphorylated (total) PKA. Without this comparative value, it becomes impossible to assess the enzyme activity. This information is lacking in Figs. 3 and 4. On what basis did the authors conclude changes in PKA activity?
  5. Please justify the choice of duloxetine as the reference drug. Many antidepressants are more commonly used in the clinic than duloxetine, so why was this drug chosen?

Reviewer 2 Report

Comments and Suggestions for Authors

General Comments:

  1. Statistical Power and Sample Size: While the statistical analyses seem appropriate (ANOVA and t-tests), the manuscript could benefit from a clearer statement of the sample sizes (n) used in each experiment (both in vitro and in vivo) within the figure legends or the methods section. Additionally, a brief discussion of the statistical power of the study to detect the observed effects would be valuable.

  2. Specificity of Compound C: Compound C is described as an "AMPK-specific inhibitor." While it is a commonly used inhibitor, it's worth briefly acknowledging in the discussion potential off-target effects, even if minimal, and the reliance on this single inhibitor for mechanistic confirmation.

  3. Translation to Humans: The study provides compelling preclinical evidence. The discussion should cautiously address the translational potential to humans. Factors such as bioavailability of UA in humans, effective brain concentrations, and potential inter-individual variability in gut microbiota metabolism of ellagitannins should be considered.

  4. Clarity of Figures: Some figure legends could be more detailed. For instance, specifying the number of replicates (biological and technical) for each experiment would enhance transparency. In Figure 2E, while the description mentions "bright blue and bright red fluorescence," explicitly stating which color corresponds to apoptosis and which to necrosis would be helpful for the reader.

Specific Comments and Areas for Improvement:

Introduction:

  1. Mechanism of Existing Antidepressants: While the introduction mentions the limitations of existing antidepressants, briefly touching upon their primary mechanisms of action (e.g., monoamine reuptake inhibition) could provide a better context for the novelty of UA's potential mechanism involving AMPK/CREB/BDNF.

  2. Blood-Brain Barrier Permeability: While the abstract mentions UA's ability to cross the blood-brain barrier, citing the specific study that demonstrated this in the introduction would be beneficial for readers seeking this foundational information early on.

Materials and Methods:

  1. Animal Welfare: While the text mentions ethical approval, adding a brief statement about adherence to specific animal welfare guidelines (e.g., AVMA guidelines) would strengthen the ethical considerations.

  2. Solvent Vehicle Control: The solvent vehicle (5% castor oil) is mentioned. It would be prudent to include data showing that the vehicle itself had no significant effects on the measured parameters in the in vivo experiments. This is often done by including a "Vehicle control" group in all relevant figures.

  3. CSDS Procedure Details: While the CSDS procedure is described, more detail on the criteria for defining "susceptible" and "resilient" mice (if applicable and if further analysis was done on subgroups) could be included for clarity.

  4. Behavioral Test Order: The order of behavioral tests in both the behavioral despair and CSDS models is mentioned in Figure 1. Briefly stating the rationale for this specific order in the methods section would be helpful, especially considering potential carry-over effects between tests.

  5. ELISA Kits: Specifying the manufacturer and catalog numbers of the ELISA kits used for ACTH and CORT measurements would enhance the reproducibility of the study.

  6. Western Blot Antibody Information: While the primary antibodies are listed, including the dilution ratios used for each antibody in the methods section would be standard practice.

Results:

  1. Quantification of Apoptosis: In Figure 2E, the description of apoptosis is qualitative ("significantly attenuated the bright blue and bright red fluorescence"). Quantifying the apoptotic cells (e.g., by counting cells with PI and DAPI staining) and presenting this data graphically would provide stronger evidence.

  2. Time Course Experiments: The study uses a single time point (24 hours for CORT, 12-hour pretreatment with UA). Investigating the time-dependent effects of UA on the AMPK/CREB/BDNF pathway could provide a more comprehensive understanding of its mechanism.

  3. Specificity of AMPK Inhibition In Vivo: While Compound C inhibited UA's effects in vivo, demonstrating the effective inhibition of p-AMPK levels in the hippocampus of Compound C-treated mice (perhaps as a supplementary figure) would further support the role of AMPK. Figure 8 shows this, which is good.

  4. Correlation Analyses: Exploring the correlation between the biochemical changes (e.g., p-AMPK, BDNF levels) and the behavioral outcomes (e.g., immobility time, sucrose preference) could strengthen the link between the molecular mechanisms and the observed antidepressant-like effects.

Discussion:

  1. Novelty and Significance: While the discussion likely highlights the novelty, explicitly stating how this study advances the understanding of UA's potential and the role of the AMPK/CREB/BDNF pathway in its effects would be beneficial.

  2. Comparison to Existing Antidepressants (Mechanism): A more in-depth comparison of the proposed mechanism of UA with those of conventional antidepressants could further emphasize its potential as a novel therapeutic agent.

  3. Limitations of the Study: The authors should explicitly acknowledge the limitations of the study, such as the use of a specific cell line, the focus on male mice, and the reliance on behavioral models that may not fully recapitulate the complexity of human depression.

  4. Future Directions: The discussion should clearly outline potential future research directions, such as investigating the effects of UA in different animal models of depression, exploring the role of specific gut microbiota in UA production and its efficacy, and conducting pharmacokinetic and pharmacodynamic studies to inform potential clinical trials. Investigating the long-term effects and potential side effects of UA would also be important.

  5. Anxiolytic-like Effects: The abstract and results mention anxiolytic-like effects in the EPMT. The discussion could elaborate further on the potential mechanisms underlying these effects, especially in relation to the AMPK/CREB/BDNF pathway or other known anxiolytic pathways.

Limits of the Study (Implicitly Present, Should be Explicitly Acknowledged):

  1. Focus on a Specific Pathway: The study strongly focuses on the AMPK/CREB/BDNF pathway. While the results are compelling, it's possible that other pathways are also involved in UA's antidepressant-like effects. Future studies could explore these.

  2. Single Sex of Mice: The in vivo studies primarily used male mice. The potential for sex-specific differences in the response to UA should be acknowledged, and future studies could include female mice.

  3. Behavioral Models: Animal models of depression have limitations in fully reflecting the heterogeneity and complexity of human depression. The findings should be interpreted within this context.

  4. Dosage and Administration: The study used specific doses of UA and a particular route of administration. Further research could explore the dose-response relationship and the effects of different administration routes.

  5. Chronic vs. Acute Effects: The study investigated the effects of UA over a specific timeframe. Examining the acute and chronic effects of UA could provide a more complete picture of its potential therapeutic utility.
